# Identification of modifier gene variants overrepresented in familial hypomagnesemia with hypercalciuria and nephrocalcinosis patients with a more aggressive renal phenotype

**Monica Vall-Palomar[1], Julieta Torchia[1], Jordi Morata[2], Monica Durán[1], Raul Tonda[2], Mireia Ferrer[3], Alex Sánchez[3,4], Gerard Cantero-Recasens[1], Gema Ariceta[1,5,6], Anna Meseguer[1,7]✪\*, Cristina Martinez[1]✪**

1 Renal Physiopathology Group, Vall d'Hebron Research Institute (VHIR), Barcelona, Spain, 2 CNAG, Centro Nacional de Análisis Genómico, Barcelona, Spain, 3 Statistics and Bioinformatic Unit (UEB), Vall d'Hebron Research Institute (VHIR), Barcelona, Spain, 4 Universitat de Barcelona, Barcelona, Spain, 5 Pediatric Nephrology Department, Vall d'Hebron University Hospital, Barcelona, Spain, 6 Pediatrics Department, School of Medicine, Universitat Autònoma de Barcelona (UAB), Cerdañola del Vallés, Spain, 7 Biochemistry and Molecular Biology Department, School of Medicine, Universitat Autònoma de Barcelona (UAB), Cerdañola del Vallés, Spain

✪ These authors contributed equally to this work.
\* ana.meseguer@vhir.org

## Abstract

Familial hypomagnesemia with hypercalciuria and nephrocalcinosis (FHHNC) is an ultra-rare autosomal recessive renal tubular disease with an incidence of <1/1.000.000 individuals, caused by loss-of-function mutations in *CLDN16* and *CLDN19*. Our study includes a unique cohort representing all known FHHNC patients in Spain, with 90% harbouring mutations in *CLDN19*. Of these, 70% carry the p.G20D mutation in homozygosis. Despite this high genetic homogeneity, our FHHNC cohort display a high phenotypic variability, even among siblings harbouring identical mutations. Patients were stratified at the extremes of the renal phenotype according to their estimated glomerular filtration rate annual decline and subjected to whole exome sequencing (WES) aiming to find candidate phenotype-modifier genes. Initial statistical analysis by SKAT-O identified numerous variants, which were then filtered based on P-value <0.01 and kidney expression. A thorough prioritization strategy was then applied by an exhaustive disease knowledge-driven exploitation of data from public databases (Human Protein Atlas, GWAS catalog, GTEx) to further refine candidate genes. Odds ratios were also calculated to identify potential risk variants. This analysis pipeline suggested several gene variants associated with a higher risk of developing a more aggressive renal phenotype. While these findings hint at the existence of genetic modifiers in FHHNC, further research is needed to confirm their role and potential clinical significance. Clinical decisions should not be based on these preliminary findings, and additional cohorts should be studied to validate and expand upon our results. This exploratory study provides a foundation for future investigations into

**Data availability statement:** The data generated in this study have been deposited and will be made findable through the European Genome-Phenome Archive (EGA) web portal (https://ega-archive.org/dacs/EGAC50000000510). Access to the data will be granted for appropriate use in research and will be governed by the provisions laid out in the terms contained in the Data Access Agreement.

**Funding:** This study was funded in part by Fondo Europeo de Desarrollo Regional (FEDER), Fondo de Investigación Sanitaria, Instituto de Salud Carlos III, Subdirección General de Investigación Sanitaria, Ministerio de Ciencia e Innovación (PI14/01107, PI18/01107, PI22/01946 to GA); by the Fundación Senefro (SEN2023 to AM); and by donations from the FHHNC patient advocacy group HIPOFAM (to GA and AM). AM research group holds the Quality Mention from the Generalitat de Catalunya (Grant No. 2021 SGR 01600 to AM). CMM is supported by the Miguel Servet program from the Instituto de Salud Carlos III, Subdirección General de Investigación Sanitaria, Ministerio de Ciencia e Innovación (CP18/00116).The funders had no role in study design, data collection and analysis, decision to publish, or preparation of the manuscript.

**Competing interests:** The authors have declared that no competing interests exist.

the genetic factors influencing FHHNC progression and may contribute to our understanding of the disease's variable expressivity potentially enabling the implementation of more tailored therapeutic strategies.

## Author summary

Familial hypomagnesemia with hypercalciuria and nephrocalcinosis (FHHNC) is an ultra-rare autosomal recessive renal tubular disease with an incidence of less than 1 in 1,000,000 individuals, caused by loss-of-function mutations in *CLDN16* and *CLDN19*. The clinical course is highly variable, with patients exhibiting unexplained phenotypic variability, even among homozygotic siblings. This paper outlines a pipeline for whole exome sequence analysis and candidate prioritization tailored to the challenges of ultra-rare diseases. Our exploratory study identifies a panel of genetic variants in potential candidate modifier genes that may contribute to understanding the risk factors associated with earlier renal failure. Given the unique challenges posed by ultra-rare diseases, achieving robust statistical power can be difficult. Therefore, these results should be interpreted with caution and not be used to guide clinical decisions until further validation in additional cohorts confirms their utility. Our findings provide a foundation for future research, offering insights into FHHNC progression and potentially informing the development of more personalized therapeutic strategies, pending further validation studies.

## Introduction

Familial hypomagnesemia with hypercalciuria and nephrocalcinosis (FHHNC) is an ultra-rare autosomal recessive renal tubular disease with an incidence of <1/1.000.000 individuals, characterized by severe urinary $Ca^{2+}$ and $Mg^{2+}$ wasting and progression to chronic kidney disease (CKD) and renal failure [1,2]. Disease loss-of-function causing mutations have been identified in the tight junction (TJ) proteins claudin 16 (CLDN16) [3] and 19 (CLDN19) [4]. Both proteins are co-expressed in the TJ of the thick ascending limb of Henle's loop and form a paracellular cation-selective pore that induces a NaCl gradient and a lumen-positive transepithelial voltage driving $Ca^{2+}$ and $Mg^{2+}$ reabsorption [5,6]. Additionally, *CLDN19* is also expressed in the retinal epithelium and a subset of FHHNC patients with *CLDN19* mutations develop severe ocular impairment [4]. No specific therapy for FHHNC exists and patients rely only on supportive treatment (high fluid intake, dietary restrictions, magnesium salts and thiazide diuretics) attempting to delay the progression to renal failure. Kidney replacement therapy remains the only curative option in end-stage renal disease patients and renal transplant is usually required during the second to third decades of life [1,2], causing a severe impairment on quality of life of patients from a very young age.

Major research efforts in the last decade have been directed towards identifying novel mutations in FHHNC patients [7] and, as a result, 73 and 24 different mutations in *CLDN16* and *CLDN19*, respectively, have been described and annotated in the Human Gene Mutation Database (HGMD) [8]. Most FHHNC patients have been found to carry *CLDN16* mutations, although in the South of Europe (mainly Spain and France) *CLDN19* mutations are more prevalent and a specific *CLDN19* so-called Hispanic founder mutation (c.59G>A; p.G20D) has been described [9,10]. Remarkably, carriers of the p.G20D mutation exhibit a prominent, but yet unexplained, phenotypic variability which is observed even between homozygotic siblings

[2,9,11,12]. This suggests the existence of other not-so-far-identified molecular events, such as phenotype modifier genes, that might contribute to the clinical variability beyond the causal genes. Here we present the first study focusing on the identification of phenotype modifier genes in a unique cohort of 45 FHHNC Spanish patients through whole exome sequencing (WES) analysis comparing extreme phenotypes, as it has been successfully applied to other complex Mendelian diseases [13–15]. Candidate modifier genes were selected by a thorough prioritization strategy combining publicly available information regarding described variants associated to disease-specific phenotypic traits from the GWAS catalog together with kidney-specific expression data from the Human Protein Atlas and Genotype Tissue Expression (GTEx) databases. Our results led to a panel of genetic variants in potential candidate modifier genes which may contribute to our understanding of the factors influencing the rate of renal failure progression in FHHNC patients (see graphical abstract in Fig 1). While these findings provide a foundation for future research and could potentially inform more personalized therapeutic approaches, they should be interpreted with caution given the inherent challenges of studying ultra-rare diseases. Further validation studies are necessary to confirm the role of these potential modifier genes in FHHNC progression.

## RESULTS

### Clinical characterization of the FHHNC cohort

The discovery cohort was composed of 30 patients from 22 unrelated families. As previously described [12], pathogenic *CLDN16* mutations were confirmed in 3 patients, while the remaining 27 patients carried *CLDN19* mutations (74% were homozygous for the p.G20D

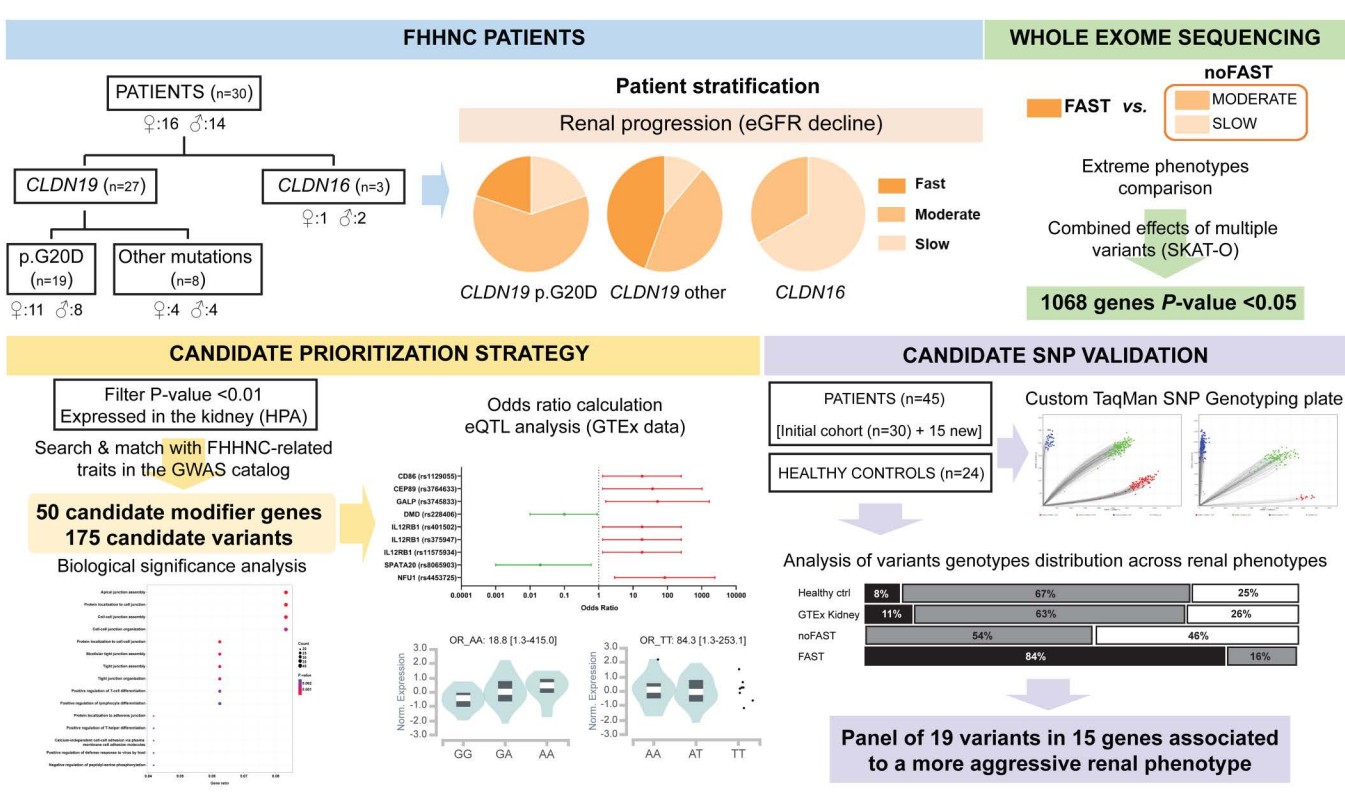

**Fig 1. Graphical abstract.**

mutation). Patients were classified based on the annual renal function decline, following the clinical criteria previously reported by our group [12].

Patients were then stratified in two extreme phenotypes for WES analysis, FAST *vs.* noFAST. In the discovery cohort (n=30), the FAST group included patients (n=6) with the fastest renal function decline, while the noFAST group (n=24) comprised patients with a moderate decline in eGFR rate and stable renal function (Fig 2 and S1 Table). Remarkably, two pairs of siblings belonging to families carrying the p.G20D in homozygosis showed different renal phenotypes (fast *vs.* moderate) and were classified in different groups for WES analysis.

During the course of this study, 15 new FHHNC patients from 14 families were recruited for validation. One of the patients carried a *CLDN16* mutation, while pathogenic mutations in *CLDN19* were confirmed in the remaining 14 patients (64% harboured the p.G20D in homozygosis). All of them were classified in the noFAST group. Additionally, we recruited 4 healthy controls among the non-affected siblings of patients, and 20 healthy controls, recruited among healthy siblings of patients with glomerulopathies from a different study, were also available.

## Whole exome sequencing

To identify candidate disease-modifying genes implicated in the phenotypic heterogeneity described in FHHNC, whole exome sequencing (WES) was performed in the discovery cohort. On average, we obtained high exon coverage of 97.22x for all 30 samples demonstrating good quality of the sequence data. Of all exons, 99.87% were covered (nonzero coverage) and 98.83% were assessed by at least 10 independent reads. Genome Analysis Toolkit (GATK) yielded a total of 110877 variants across the different samples. WES results confirmed the genetic diagnostic previously defined by Sanger sequencing for each patient. Optimal sequence kernel association test (SKAT-O) statistics were calculated using called variants with moderate and high effect types (29046 variants mapping to 11481 genes, 26.2%) according to snpEff predicted impact (missense, splice region, splice donor, splice acceptor, stop-loss, stop-gain) which yielded 1068 significant genes potentially modifying the phenotype severity (P<0.05). Functional profiling analysis revealed that these genes were enriched in Gene Ontology (GO)-Biological Processes (BP) terms related to cell-cell adhesion, response to pro-inflammatory signals (interleukin 1) and tissue regeneration; and in GO-Cellular Component (CC) terms related to plasma membrane (S1 Fig).

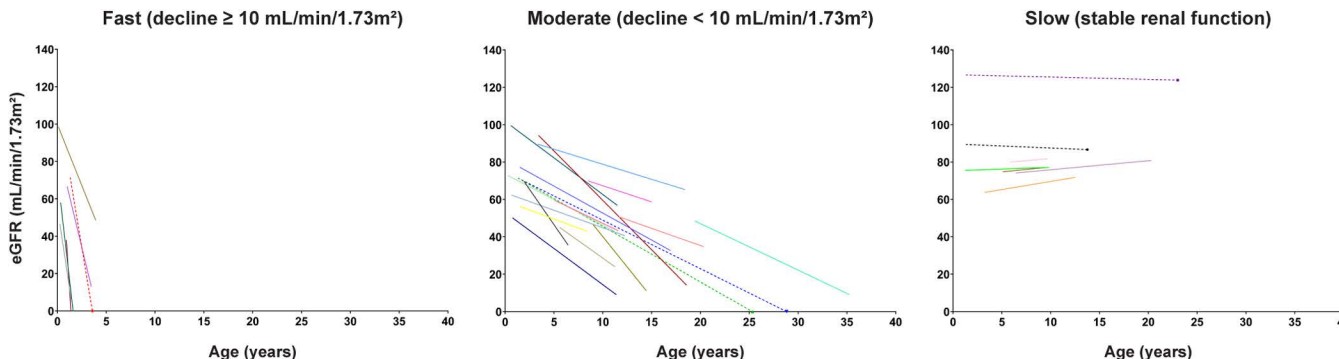

**Fig 2. Annual renal function decline.** Patterns of eGFR change over time were plotted for each patient and stratified as fast, moderate or slow according to the curve slopes determined by linear regression. Each coloured line indicates the annual eGFR loss. Dashed lines indicate patients for whom we only had one value of eGFR corresponding to the last follow-up.

## Candidate gene prioritization identifies variants associated to a faster progression to end-stage renal disease in FHHNC patients

To provide a clear overview of our gene prioritization approach, we have developed a flow diagram (Fig 3) that illustrates the step-by-step process used to identify potential modifier genes associated with FHHNC progression.

First, the SKAT-O genes were filtered by selecting those with a P-value <0.01 and described to be expressed in the human kidney in the Human Protein Atlas database, at both RNA and

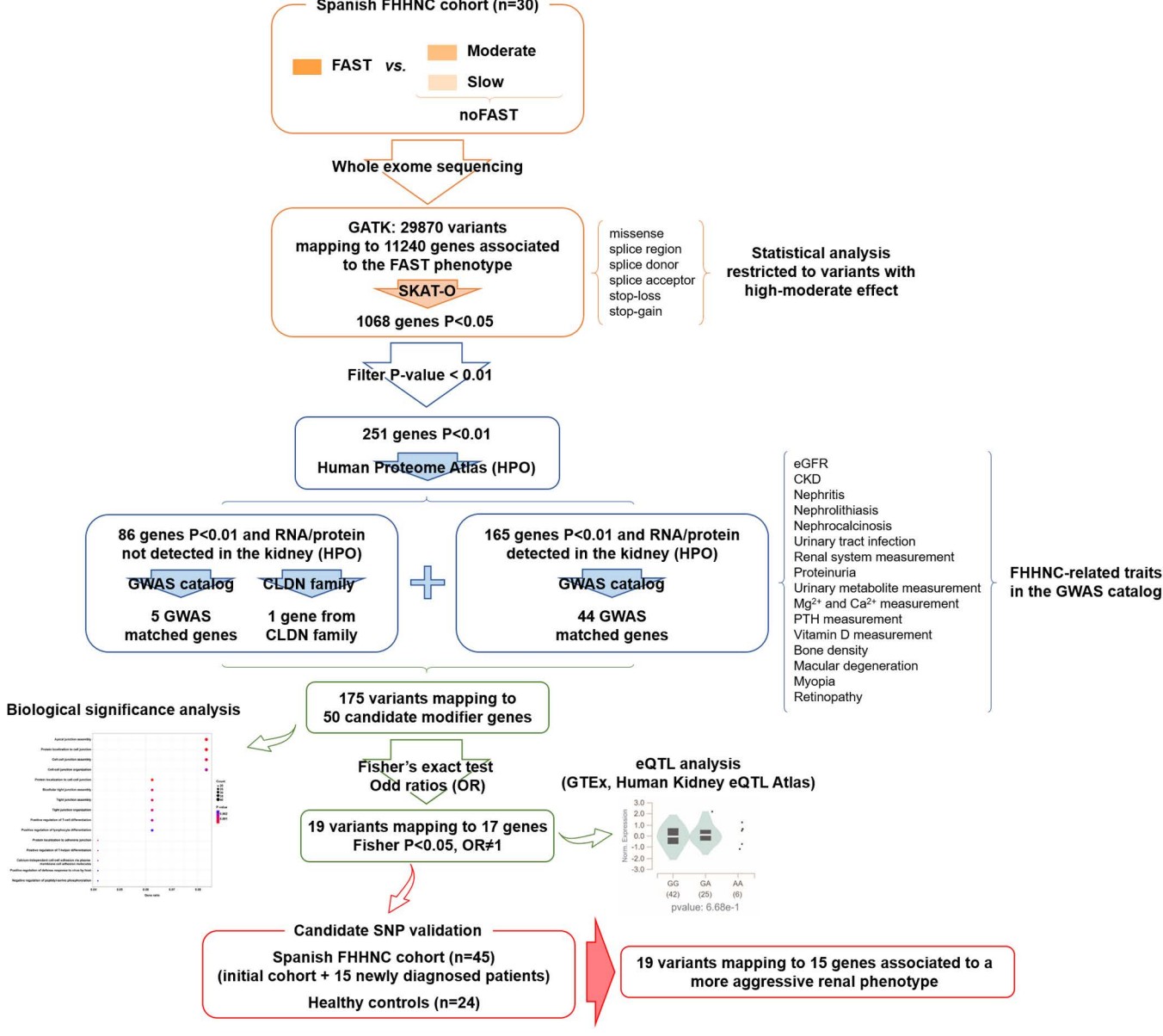

**Fig 3. Prioritization strategy.** Flow diagram illustrating the step-by-step process used to identify potential modifier genes associated with FHHNC progression. CKD, chronic kidney disease; CLDN, claudin; eGFR, estimated glomerular filtration rate; eQTL, expression quantitative trait loci; FHHNC, familial hypomagnesemia with hypercalciuria and nephrocalcinosis; GATK, Genome Analysis Toolkit; GWAS, genome wide association study; PTH, parathyroid hormone; SKAT-O, optimal sequence kernel association test.

protein evidence level. These filters reduced the candidate gene list to 165 genes. As a second step to further prioritize potential modifier genes, we searched the GWAS catalog database to identify gene variants associated to FHHNC specific phenotypic traits. Those specific traits were selected based on the renal phenotype described in our patient cohort (annual glomerular filtration rate decline) plus the wide range of common FHHNC symptoms which include polyuria-polydipsia, failure to thrive, recurrent urinary tract infections, hypercalciuria, nephrocalcinosis, hypomagnesemia, and hyperparathyroidism [2,16,17]. In addition, we also included traits related to the ocular impairment described for some carriers of *CLDN19* mutations. S2 Table summarizes the selected phenotypic traits, the number of associated variants found in the GWAS catalog and the number of matched genes from our SKAT-O analysis after filtering by P<0.01 and kidney expression.

As a result, the list of potential candidates for further analysis was reduced to 44 candidate modifier genes. We performed a second round of data matching between SKAT-O genes and GWAS catalog data including genes not expressed in the kidney in order to not exclude the possibility that genes from other tissue and/or cell types may have an impact on FHHNC progression. This second data-matching analysis identified only another 5 candidates. Additionally, as the claudin family of proteins are highly involved in the physiopathology of FHHNC, we included in the list of selected candidates the gene *CLDN17* that was significantly associated (P<0.01) to the FAST phenotype. Applying the described strategy for candidate prioritization, we finally selected 175 variants mapping to 50 genes for further analysis (Table 1).

To uncover the biological significance of the prioritized genes, an overrepresentation analysis (ORA) was performed using the GO-BP followed by clustering analysis to summarize the results obtained (summORA). Among the top 15 terms identified by ORA, those related to apical junction regulation and assembly, cell adhesion and immune-related terms were the most significant biological processes associated to our 50 phenotype modifier candidates (Fig 4A). After clustering by similarity the enriched GO-BP terms obtained with an adjusted P-value < 0.25, 12 clusters were identified (Fig 4B). Both the most representative term found in each cluster and the 10 most frequent words within them, highlighted apical junction regulation and assembly, immune-related terms, ionic transport, protein modifications (mainly phosphorylation), cellular responses to different stimulus, and cell cycle regulation as the most significant biological processes enriched in the FHHNC prioritized candidate modifier genes.

Since SKAT-O does not provide any information regarding whether the candidate genes confer a risk or a protective effect, we took a complementing approach by performing Fisher's exact test and calculating odds ratios for each individual variant identified in the SKAT-O prioritized candidate genes. Ten single nucleotide variants (SNVs) in homozygosis and nine in heterozygosis were significantly associated to renal progression phenotypes (Fisher P-value <0.05 and FDR adjusted P-value <0.15; Odds ratio ≠1) considering the alternative allele both in homozygosis and in heterozygosis (Fig 5 and S3 Table). From those, the alternative allele in homozygosis for SNVs mapping to genes *CD86*, *CEP89*, *GALP*, *IL12RB1*, *NFU1* and *SLC17A1* was associated to the FAST phenotype; while rs228406 (*DMD*) and rs8065903 (*SPATA20*) were associated to the noFAST phenotype (Fig 5A). Additionally, the alternative allele in heterozygosis for SNVs mapping to genes *AGBL2*, *ANKLE2*, *CLDN17*, *CLDN8*, *DMD*, *HSP5*, *IL12RB1*, *ITGA1* and *ZNF607* was associated to the FAST phenotype (Fig 5B). Variants in these genes are annotated in the GWAS catalog associated to $Ca^{2+}$ levels (*CD86, NFU1, AGBL2*), eGFR (*CEP89, IL12RB1, SPATA20, ZNF607*), blood urea nitrogen levels (*DMD, HPS5, ZNF607*), microalbuminuria (*GALP*), bone mineral density (*ITGA1*) and high myopia (*ANKLE2*) (Table 1). In addition, most of these genes were found in clusters related to immune system activity and regulation of apical junctions in the summORA analysis (S4 Table).

Table 1. List of 50 prioritized candidate modifier genes.

| Gene ID | SKATO P-value | SKATO Adj.P-value | Kidney expression | Variants found in FHHNC | Variants found in GWAS catalog | GWAS catalog main trait |
|---|---|---|---|---|---|---|
| OSBPL6 | 0.0002 | 0.31 | Yes | 3 | 1 | High myopia |
| HLCS | 0.0004 | 0.31 | Yes | 3 | 1 | eGFR |
| NFU1 | 0.0006 | 0.31 | Yes | 1 | 1 | $Ca^{2+}$ |
| NINJ1 | 0.0008 | 0.31 | Yes | 1 | 2 | eGFR |
| DMD | 0.001 | 0.31 | Yes | 5 | 1 | Renal system measurement |
| IL12RB1 | 0.002 | 0.31 | Yes | 5 | 2 | eGFR |
| ANKLE2 | 0.002 | 0.31 | Yes | 7 | 1 | High myopia |
| DOCK9 | 0.002 | 0.31 | Yes | 4 | 6 | Heel bone mineral density/ low myopia |
| CEP89 | 0.003 | 0.31 | Yes | 5 | 5 | eGFR |
| MAST2 | 0.003 | 0.31 | Yes | 7 | 2 | Serum creatinine/ Serum 25-Hydroxyvitamin D |
| HPS5 | 0.003 | 0.31 | Yes | 2 | 1 | Renal system measurement |
| CLDN17 | 0.003 | 0.31 | No | 1 | 0 | NA |
| SIN3A | 0.003 | 0.31 | Yes | 1 | 4 | eGFR |
| RGS19 | 0.003 | 0.31 | Yes | 1 | 1 | eGFR |
| CMAHP | 0.003 | 0.31 | No | 1 | 1 | Creatinine in ischemic stroke |
| TXNL1 | 0.003 | 0.31 | Yes | 1 | 1 | 6-month creatinine clearance change response to tenofovir in HIV |
| SLC35F1 | 0.003 | 0.31 | No | 1 | 1 | Heel bone mineral density |
| GRAMD1A | 0.003 | 0.31 | Yes | 1 | 1 | $Ca^{2+}$ |
| EIF1AD | 0.003 | 0.31 | Yes | 1 | 1 | Renal System Measurement |
| GLI2 | 0.003 | 0.31 | Yes | 3 | 1 | eGFR in coronary artery disease and impaired kidney function |
| SPG7 | 0.003 | 0.31 | Yes | 3 | 2 | Heel bone mineral density |
| HHAT | 0.003 | 0.31 | Yes | 4 | 1 | High myopia |
| MICALL1 | 0.004 | 0.35 | Yes | 3 | 2 | Heel bone mineral density |
| PLEKHG1 | 0.005 | 0.37 | Yes | 4 | 2 | Retinal drusen |
| PMFBP1 | 0.005 | 0.37 | Yes | 4 | 3 | Urinary metabolite measurement |
| SLC9A3 | 0.005 | 0.37 | Yes | 1 | 2 | eGFR |
| GALP | 0.005 | 0.37 | No | 1 | 1 | Microalbuminuria |
| CCDC57 | 0.005 | 0.38 | Yes | 10 | 1 | Renal system measurement |
| RHPN2 | 0.005 | 0.38 | Yes | 4 | 12 | Bone mineral density/ macular thickness |
| UNC13C | 0.006 | 0.38 | No | 2 | 2 | Retinal drusen/ proteinuria in preeclampsia |
| RBL2 | 0.006 | 0.39 | Yes | 5 | 1 | eGFR |
| NADSYN1 | 0.006 | 0.39 | Yes | 3 | 14 | Vitamin D |
| ITGA1 | 0.006 | 0.39 | Yes | 5 | 2 | Heel bone mineral density |
| PITPNM2 | 0.008 | 0.45 | Yes | 2 | 2 | $Ca^{2+}$/ macular thickness |
| SORT1 | 0.008 | 0.45 | Yes | 3 | 2 | eGFR |
| DLG5 | 0.008 | 0.45 | Yes | 5 | 1 | Retinal detachment or retinal break/ macular thickness |
| AGBL2 | 0.008 | 0.45 | Yes | 2 | 1 | $Ca^{2+}$ |
| CGNL1 | 0.008 | 0.45 | Yes | 5 | 14 | eGFR/ macular thickness |
| CACNA1S | 0.008 | 0.47 | No | 8 | 2 | eGFR |
| MLXIPL | 0.008 | 0.47 | Yes | 4 | 8 | Renal system measurement/ $Ca^{2+}$ |
| MADD | 0.009 | 0.47 | Yes | 4 | 1 | CKD |
| NPHS1 | 0.009 | 0.47 | Yes | 5 | 3 | eGFR/ Serum 25-Hydroxyvitamin D |
| ZNF607 | 0.009 | 0.48 | Yes | 3 | 3 | eGFR/ Renal system measurement |
| GPRIN3 | 0.01 | 0.49 | Yes | 6 | 1 | Urinary $Ca^{2+}$ excretion |
| CD86 | 0.01 | 0.50 | Yes | 1 | 8 | $Ca^{2+}$ |
| SPATA20 | 0.01 | 0.50 | Yes | 3 | 3 | eGFR/ bone mineral density |

*(Continued)*

**Table 1.** (Continued)

| Gene ID | SKATO P-value | SKATO Adj.P-value | Kidney expression | Variants found in FHHNC | Variants found in GWAS catalog | GWAS catalog main trait |
|---------|------|------|-----|----|----|-----------------------------------|
| SLC17A1 | 0.01 | 0.50 | Yes | 1 | 16 | Renal system measurement |
| MPP7 | 0.01 | 0.50 | Yes | 1 | 14 | Bone mineral density/ macular thickness |
| CLDN8 | 0.01 | 0.50 | Yes | 2 | 1 | Renal system measurement |
| PLIN4 | 0.01 | 0.50 | Yes | 17 | 2 | eGFR |

**Notes:** Ca$^{2+}$, calcium levels; CKD, chronic kidney disease; eGFR, estimated glomerular filtration rate.

### Identified phenotype modifier variants are overrepresented in FHHNC patients with a faster renal function decline compared to patients with moderate/mild phenotype and healthy individuals

In order to understand the functional implications of SNVs associated to renal progression in FHHNC, we retrieved expression quantitative trait loci (eQTL) analysis data for each significant variant in 659 microdissected (309 glomeruli and 350 tubules) human kidney samples from the Human Kidney eQTL Atlas [30] and in human kidney cortex tissues (n=73) from the GTEx database. Variant rs4453725 for *NFU1*, which we have identified as a risk factor for eGFR decline, was found to be a significant eQTL in microdissected renal tubules from the Human Kidney eQTL Atlas (P-value = 2.14$^{-12}$). Conversely, analysis of whole renal cortex tissue from GTEx revealed no significant impact of the identified gene variants on gene expression, suggesting that the impact of *NFU1* variant on disease progression may be mediated through gene expression changes restricted to renal tubules, rather than through broad alterations in overall kidney gene expression.

Notably, the risk alleles for SNVs associated with rapid renal progression were significantly underrepresented in GTEx kidney tissues (Fig 6 and S5 Table). Furthermore, the FAST renal progression phenotype showed a markedly higher proportion of patients carrying these risk alleles compared to both the noFAST group and the expected frequency in the general healthy population, as represented by control tissues from GTEx (Fig 7). Interestingly, not all initially prioritized variants were found to be overrepresented in the FAST cohort. This discrepancy may indicate that some of the initially prioritized variants could have been false positives. Nonetheless, this analysis has allowed us to further refine the candidate variants, narrowing down those consistently associated with faster progression of renal failure across different analytical approaches.

Next, to validate our results, the 19 variants associated to a faster renal progression were genotyped in a second cohort. To do this, 15 additional FHHNC patients were recruited together with a healthy control cohort. However, none of the new patients met the criteria to be classified in the FAST group. Therefore, and to maximize the study power, we genotyped the full cohort of patients using a customized OpenArray panel (n=45). Our data showed that, in the FAST group the distribution of the three possible genotypes for each variant were identical to the data obtained by WES, while adding 15 patients to the noFAST group rendered highly similar results to those in the WES analysis (Table 2). Moreover, the healthy control cohort showed also a genotype distribution similar to those from the general healthy population (GTEx).

Next, we aimed to identify whether the prioritized SNVs showed a combined effect on the renal phenotype. Unfortunately, polygenic risk scores could not be determined as these SNVs are not annotated in any genome-wide study for FHHNC-related traits. However, most of the

**A.**

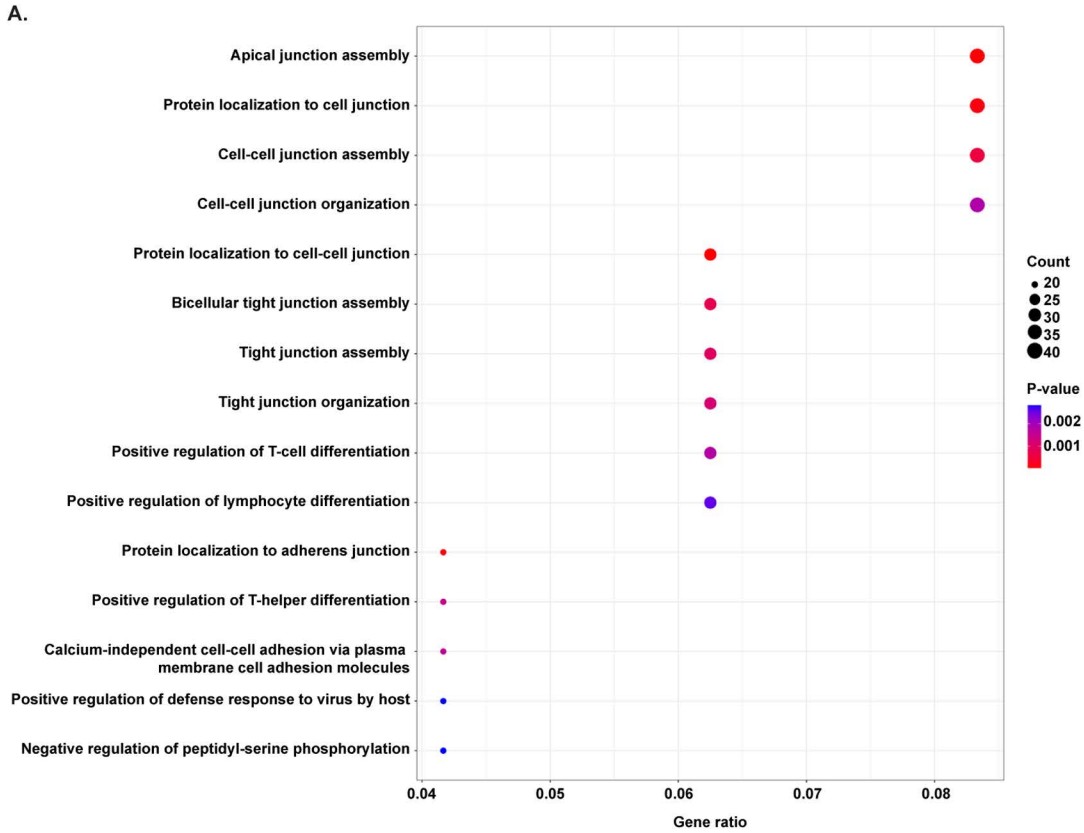

**B.**

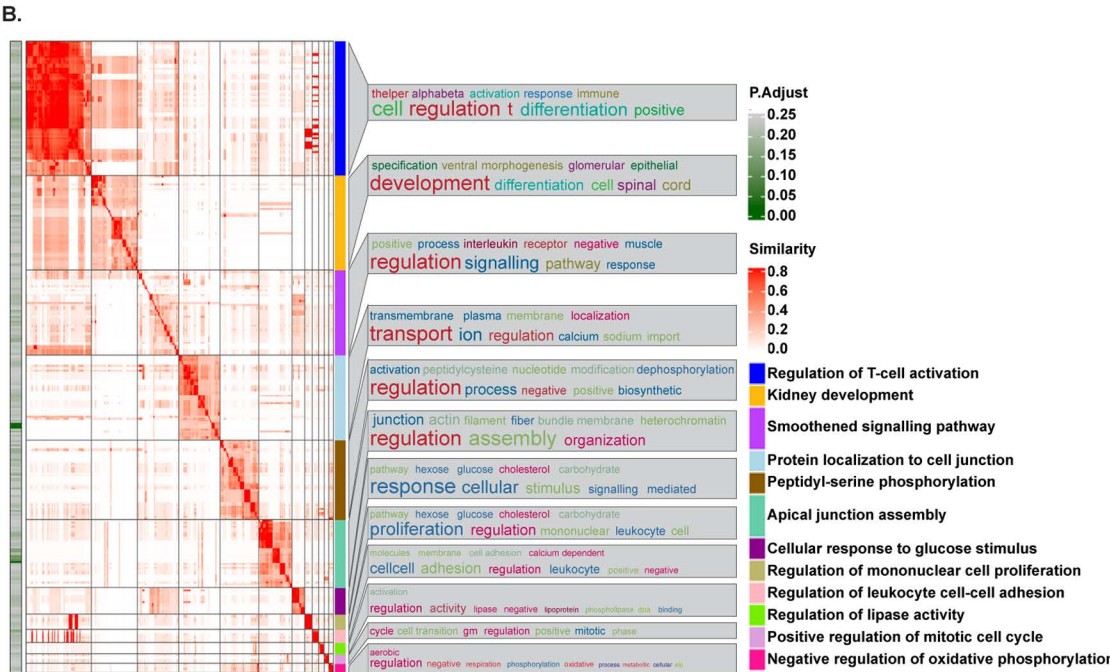

**Fig 4. Overrepresentation analysis of the 50 prioritized candidate modifier genes in FHHNC patients. A.** Dot plot showing the top 15 most significant biological processes overrepresented. The size of the dots relates to the number of genes/proteins in the data that belong to that pathway, the colour of the plot refers to significance level (P-value). The terms are ordered by P-value and Gene ratio, which is the ratio between the genes/proteins in the data that belong to that term and the total number of genes/proteins in the term. **B.** Heatmap of clustering analysis summarizing overrepresentation results. Clusters are annotated with the most

representative term within each cluster (the term with the lowest P-value and the higher number of candidate genes) and with a word cloud representing the top 10 most frequent words found within term names in each cluster. The heatmap shows the terms found within each list, coloured according to their adjusted P-value (P.Adjust).

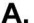

**A.**

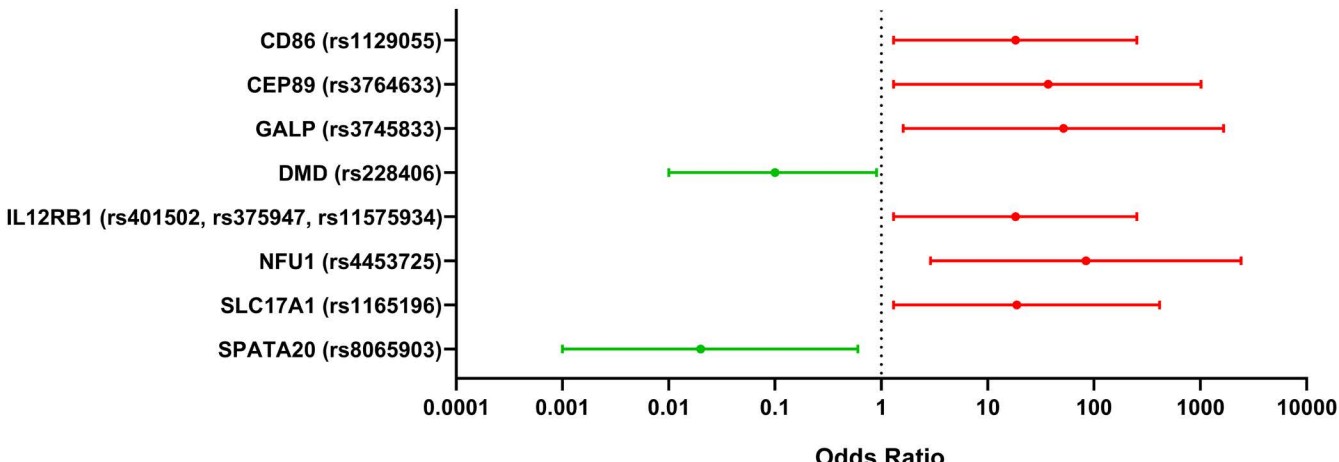

**B.**

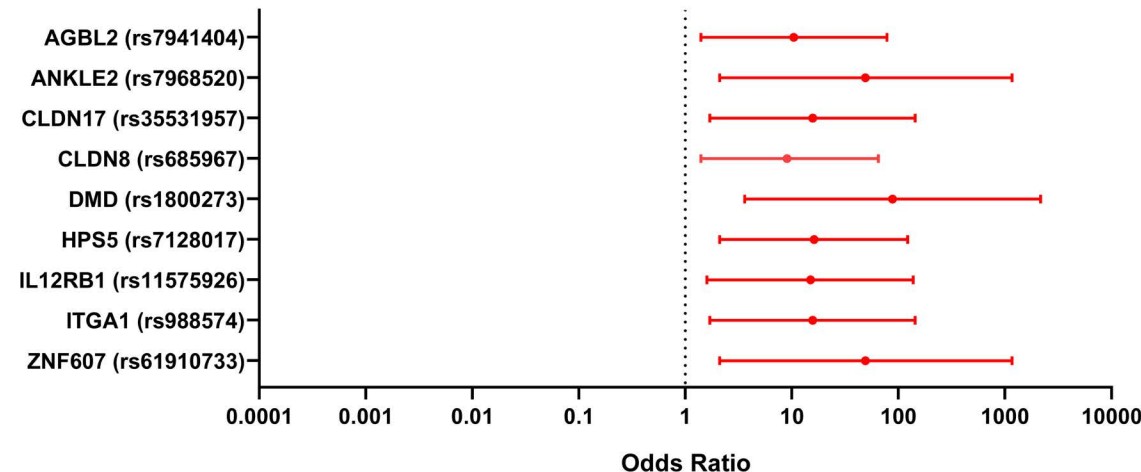

**Fig 5. Forest plot showing the significant variants in candidate modifier genes associated to the FAST phenotype of FHHNC patients. A.** Odds ratios and 95% confidence intervals were calculated by comparing the distribution of the alternative allele in homozygosis *versus* the reference allele in homozygosis. **B.** Odds ratios and 95% confidence intervals were calculated by comparing the distribution of the alternative allele in heterozygosis *versus* the reference allele in homozygosis.

patients in the FAST group (67%) carried between 9 and 15 SNVs in the candidate modifier genes, while the maximum number of SNVs found in the noFAST group was 4 (S6 Table). This distribution highlights a potential cumulative impact of these SNVs on the progression of renal failure in the FAST cohort.

## Discussion

The FHHNC clinical course is highly variable, whereas some individuals require kidney replacement therapy early in childhood, others maintain a more stable renal function into

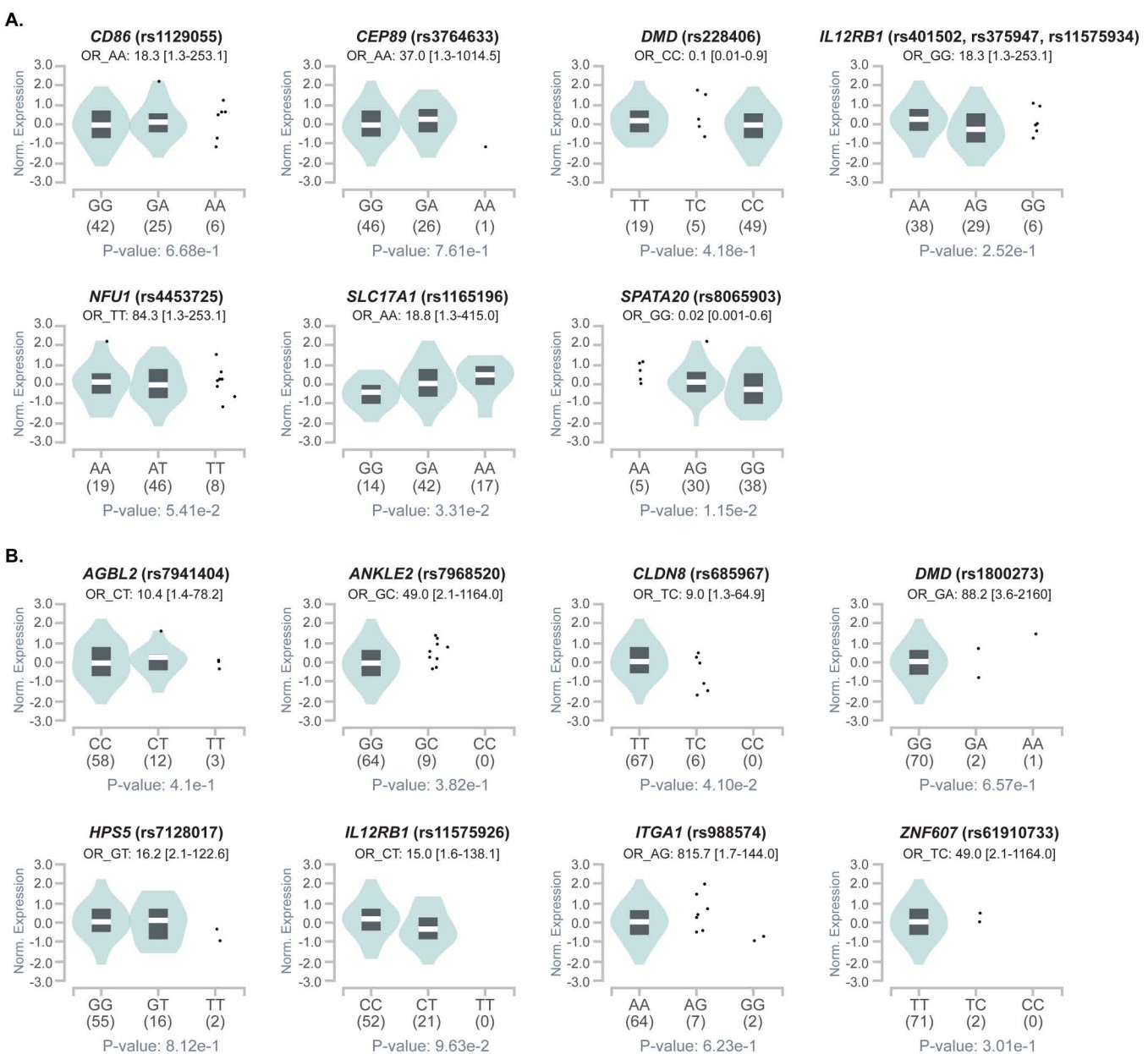

**Fig 6. Violin plots of allele-specific cis-eQTLs according to genotypes for each significant variant in human kidney cortex tissues from the GTEx database.** The teal region indicates the density distribution of the samples in each genotype. The allelic effect of variants on normalized gene expression levels are shown by boxplots within violin plots. The white line in the box plot (black) shows the median value of the expression of each genotype. The alleles and number of subjects analysed is shown under each genotype. **A.** Variants in homozygosis for the alternative allele associated to fast renal progression. **B.** Variants in heterozygosis associated to fast renal progression. The odds ratio and 95% confidence intervals calculated from Table 2 are shown next to each plot for reference.

adulthood suffering with only mild to moderate renal disease [12]. Major research efforts in the field are focused on identifying novel mutations in the known causing genes, *CLDN16* and *CLDN19*, while other potential genetic factors underlying phenotypic variability within patients have been ignored, likely due to the difficulties inherent to performing exome-wide association approaches in an ultra-rare disease. In the exploratory study presented here we

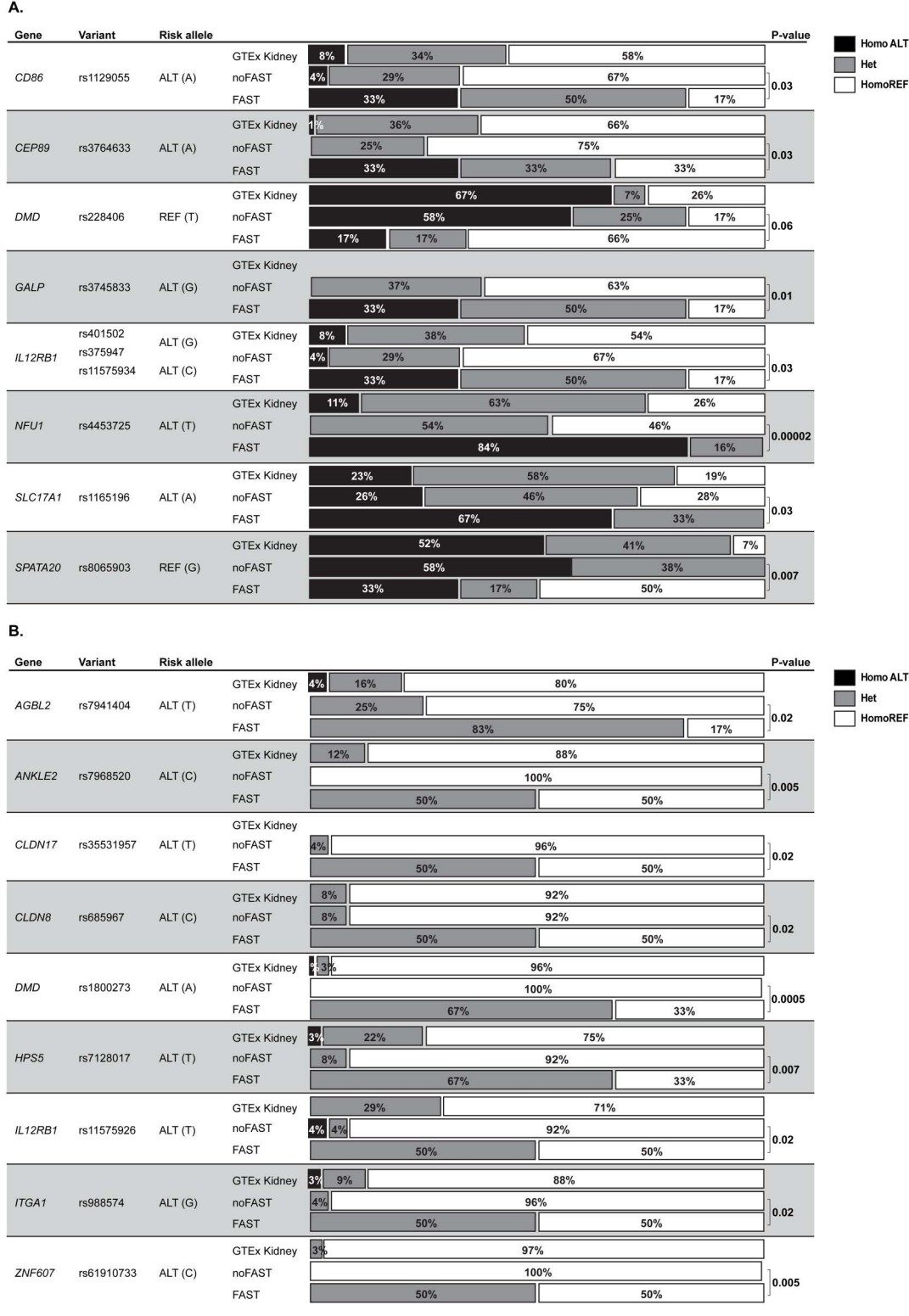

**Fig 7. Distribution of the three possible genotypes for each candidate modifier gene variants across two FHHNC phenotypes and the control group from GTEx. A.** Variants in homozygosis for the alternative allele associated to fast renal progression. **B.** Variants in heterozygosis associated to fast renal progression. Comparisons were performed by the Fisher's exact test.

**Table 2.** Comparison of the genotype distribution obtained by Open Array genotyping data in the validation cohort and by whole exome sequencing data from the discovery cohort. Data from the three possible genotypes is shown for each candidate modifier gene variant across the two FHHNC renal phenotypes (FAST and noFAST), the healthy control cohort and the control group from GTEx.

| | Gene Name | Variant ID | Risk allele | Genotype | FAST (%) | | NoFAST (%) | | Healthy Ctrl (%) | GTEx (%) |
|---|---|---|---|---|---|---|---|---|---|---|
| | | | | | WES (n=6) | OA SNP (n=6) | WES (n=24) | OA SNP (n=39) | OA SNP (n=24) | (n=73) |
| **Homozigous alternative allele** | *CD86* | rs1129055 | ALT (A) | HomALT | 33% | 33% | 4% | 5% | 4% | 8% |
| | | | | HomREF | 17% | 17% | 67% | 67% | 58% | 58% |
| | | | | Het | 50% | 50% | 29% | 28% | 38% | 34% |
| | *CEP89* | rs3764633 | ALT (A) | HomALT | 33% | 33% | 0% | 3% | 4% | 1% |
| | | | | HomREF | 33% | 33% | 75% | 79% | 66% | 66% |
| | | | | Het | 33% | 33% | 25% | 18% | 30% | 36% |
| | *DMD* | rs228406 | REF (T) | HomALT | 17% | na | 58% | na | na | 67% |
| | | | | HomREF | 66% | na | 17% | na | na | 26% |
| | | | | Het | 17% | na | 25% | na | na | 7% |
| | *GALP* | rs3745833 | ALT (G) | HomALT | 33% | 33% | 0% | 8% | 17% | na |
| | | | | HomREF | 17% | 17% | 63% | 51% | 25% | na |
| | | | | Het | 50% | 50% | 37% | 41% | 58% | na |
| | *IL12RB1* | rs401502 | ALT (G) | HomALT | 33% | 33% | 4% | 3% | 4% | 8% |
| | | | | HomREF | 17% | 17% | 67% | 62% | 63% | 54% |
| | | | | Het | 50% | 50% | 29% | 35% | 33% | 38% |
| | | rs375947 | ALT (G) | HomALT | 33% | 36% | 4% | 4% | 4% | 8% |
| | | | | HomREF | 17% | 18% | 67% | 60% | 63% | 54% |
| | | | | Het | 50% | 45% | 29% | 36% | 33% | 38% |
| | | rs11575934 | ALT (C) | HomALT | 33% | na | 4% | na | na | 8% |
| | | | | HomREF | 17% | na | 67% | na | na | 54% |
| | | | | Het | 50% | na | 29% | na | na | 38% |
| | *NFU1* | rs4453725 | ALT (T) | HomALT | 84% | 83% | 0% | 5% | 8% | 11% |
| | | | | HomREF | 16% | 17% | 46% | 47% | 25% | 26% |
| | | | | Het | 0% | 0% | 54% | 48% | 67% | 63% |
| | *SLC17A1* | rs1165196 | ALT (A) | HomALT | 67% | 67% | 21% | 26% | 25% | 23% |
| | | | | HomREF | 0% | 0% | 33% | 28% | 17% | 19% |
| | | | | Het | 33% | 33% | 46% | 46% | 58% | 58% |
| | *SPATA20* | rs8065903 | REF (G) | HomALT | 33% | 33% | 58% | 63% | 67% | 52% |
| | | | | HomREF | 50% | 50% | 0% | 0% | 0% | 7% |
| | | | | Het | 17% | 17% | 38% | 37% | 33% | 41% |
| **Heterozigous** | *AGBL2* | rs7941404 | ALT (T) | HomALT | 0% | 0% | 0% | 0% | 8% | 4% |
| | | | | HomREF | 17% | 17% | 75% | 74% | 79% | 80% |
| | | | | Het | 83% | 83% | 25% | 26% | 13% | 16% |
| | *ANKLE2* | rs7968520 | ALT (C) | HomALT | 0% | 0% | 0% | 0% | 0% | 0% |
| | | | | HomREF | 50% | 50% | 100% | 92% | 88% | 88% |
| | | | | Het | 50% | 50% | 0% | 8% | 13% | 12% |
| | *CLDN17* | rs35531957 | ALT (T) | HomALT | 0% | 0% | 0% | 0% | 0% | na |
| | | | | HomREF | 50% | 50% | 96% | 92% | 79% | na |
| | | | | Het | 50% | 50% | 4% | 8% | 21% | na |
| | *CLDN8* | rs685967 | ALT (C) | HomALT | 0% | 0% | 0% | 0% | 0% | 0% |
| | | | | HomREF | 50% | 50% | 92% | 87% | 88% | 92% |
| | | | | Het | 50% | 50% | 8% | 13% | 12% | 8% |

*(Continued)*

**Table 2.** (Continued)

| | Gene Name | Variant ID | Risk allele | Genotype | FAST (%) | | NoFAST (%) | | Healthy Ctrl (%) | GTEx (%) |
|---|---|---|---|---|---|---|---|---|---|---|
| | | | | | WES (n=6) | OA SNP (n=6) | WES (n=24) | OA SNP (n=39) | OA SNP (n=24) | (n=73) |
| | *DMD* | rs1800273 | ALT (A) | HomALT | 0% | 0% | 0% | 5% | 4% | 1% |
| | | | | HomREF | 33% | 33% | 100% | 95% | 91% | 96% |
| | | | | Het | 67% | 67% | 0% | 0% | 4% | 3% |
| | *HPS5* | rs7128017 | ALT (T) | HomALT | 0% | 0% | 0% | 0% | 0% | 3% |
| | | | | HomREF | 33% | 33% | 92% | 74% | 72% | 75% |
| | | | | Het | 67% | 67% | 8% | 26% | 28% | 22% |
| | *IL12RB1* | rs11575926 | ALT (T) | HomALT | 0% | 0% | 4% | 4% | 0% | 0% |
| | | | | HomREF | 50% | 50% | 92% | 79% | 83% | 71% |
| | | | | Het | 50% | 50% | 4% | 17% | 17% | 22% |
| | *ITGA1* | rs988574 | ALT (G) | HomALT | 0% | 0% | 0% | 0% | 0% | 3% |
| | | | | HomREF | 50% | 50% | 96% | 97% | 83% | 88% |
| | | | | Het | 50% | 50% | 4% | 3% | 17% | 9% |
| | *ZNF607* | rs61910733 | ALT (C) | HomALT | 0% | 0% | 0% | 0% | 0% | 0% |
| | | | | HomREF | 50% | 50% | 100% | 100% | 96% | 97% |
| | | | | Het | 50% | 50% | 0% | 0% | 4% | 3% |

**Notes:** ALT, alternative allele; REF, reference allele; WES, whole exome sequencing; OA, open array; na, not applicable.

describe the pipeline for WES analysis and candidate prioritization that has allowed us to identify potential phenotype modifier gene variants associated to a faster renal progression in FHHNC patients.

Our study was specifically designed to maximise the chances of identifying genes with significant associations in a small patient cohort, which benefits from the following key features. First, the unique cohort encompassing nearly all FHHNC patients diagnosed in Spain prior to January 2021, with detailed clinical and genetic characterization. Second, the high genetic homogeneity of our cohort with 90% of patients harbouring *CLDN19* mutations, including 70% with the homozygous p.G20D founder mutation. Third, we employed an extreme phenotype study design based on the assumption that risk variants would segregate with a more severe phenotype. This approach has been successfully applied to other diseases and has been proven to increase power, particularly when dealing with limited sample sizes as in rare diseases [14,15]. Therefore, we stratified our patients at the extremes of a quantitative trait as the annual decline on the estimated glomerular filtration rate as previously reported by our group [12]. And fourth, we applied a WES data analysis pipeline assessing mixed effects of variants using SKAT-O. SKAT-O is a unified approach that optimally combines the burden and non-burden sequence kernel association test (SKAT) and increases statistical power to detect associations, particularly when used in combination with extreme phenotypes analysis strategy [18].

By using SKAT-O and comparing patients with a fast renal progression to patients with a moderate/mild renal progression, we obtained a high number of genes associated to the most severe renal phenotype. As expected, due to the limited number of samples analysed, none of those candidate genes resisted multiple testing correction. To mitigate the risk of false positives, we implemented a prioritization strategy focused on selecting genes relevant to FHHNC based on current state-of-the-art knowledge of the disease. This strategy consisted

on i) reducing our threshold for significance to P<0.01, ii) filtering the list for those genes expressed in renal tissue, and iii) further filtering the list selecting genes that had been previously associated to FHHNC key phenotypic traits by GWAS. Remarkably, overrepresentation analysis on prioritized candidates identified apical junction regulation and assembly, cell adhesion and immune system-related processes as the most significant terms associated to the modifier variants in the fast renal progression group which indicates that our analysis pipeline is selecting the candidates that are better aligned to the physiopathological hallmarks of the disease and, thus, most likely to have a true involvement in its clinical course. We acknowledge, however, that the extra filtering steps by tissue expression and GWAS annotation may be suspicious of introducing some bias in the enrichment analysis (terms related to kidney or to the traits used for GWAS-annotation would probably be favoured due to the pre-filtering). Nonetheless, it should be noted that the overrepresentation analysis has been applied only as means to functionally annotate the selected genes and it did not have any influence on the candidate prioritization strategy.

Taking all the above considerations together, we took a further step to minimize the chances of selecting false positives and to prioritize gene variants associated to a higher risk of a faster renal progression by calculating the Fisher's exact test and odds ratios. This additional analysis reduced the prioritized genes to 19 variants in 15 genes associated to a faster renal progression. Variants in these genes are annotated in the GWAS catalog associated to $Ca^{2+}$ levels, eGFR, blood urea nitrogen levels, microalbuminuria, bone mineral density and high myopia, although the precise mechanisms underlying these associations have not been explored. Moreover, the variants found in our FHHNC cohort are different from those annotated in the GWAS catalog and their involvement in renal failure progression remains to be elucidated. Interestingly, among the genes conferring increased risk to faster renal progression, the biggest effect size corresponded to the homozygotic variant p.M25K in gene *NFU1*. *NFU1* is involved in iron–sulphur [Fe-S] cluster assembly, a highly complex system involved in several cellular processes including mitochondrial respiratory chain activity and various other enzymatic and regulatory functions [19–21]. Defects in this process have a severe impact on human health [19,22] and, indeed, loss-of-function mutations in *NFU1* and other proteins involved in [Fe-S] cluster assembly have been described in multiple mitochondrial dysfunction syndromes (MMDS), a rare disease characterized by neurological regression, reduced motor control (dystonia) and pulmonary hypertension. Although almost all *NFU1* pathogenic mutations described so far are located in the C-terminal NifU [Fe-S] cluster binding domain [23], the *NFU1* p.R21P mutation located in the N-domain, which is only 4 aa apart from p.M25K and affects the mitochondrial target signal [24], has been described in the only MMDS patient showing renal tubular impairment [23,25]. Renal involvement has been also described for MMDS patients harbouring mutations in *BOLA3* [23], a protein that cooperates with *NFU1* in the last steps of maturation and transfer of [4Fe-4S] clusters into target proteins, further supporting the role of [Fe-S] cluster assembly defects on kidney disease. Based on our data, we suggest that [Fe-S] cluster homeostasis may be imbalanced in renal tubular cells from FHHNC patients with a more severe renal phenotype contributing, therefore, to the aggravation of the high cellular stress already produced by the *CLDN19* mutation. The potential involvement of the p.M25K *NFU1* variant in FHHNC renal failure progression is further supported by the fact that it is considered a common variant with a MAF of 0.40 according to the gnomAD browser. Therefore, its potential impact on cellular homeostasis is likely to be easily accommodated in healthy individuals, particularly when the variant is present in heterozygosis, as it happens in the majority of cases genotyped. Indeed, the GTEx data showed that the p.M25K *NFU1* variant was present in homozygosis in 11% of the genotyped renal tissues and in 8% of the healthy subjects in our study. Remarkably, our multi-step analytical approach

revealed that only 5% of FHHNC patients (2 out of 39) with a moderate/mild progression had this variant in homozygosis, while it was present in 5 out of 6 patients with a fast renal progression.

This overrepresentation in patients with a more severe phenotype was not unique to the *NFU1* variant but was a common observation for almost all the other variants prioritized in our study. Of particular interest are the variants found in the genes encoding dystrophin (*DMD*) and Hermansky-Pudlak Syndrome protein 5 (*HPS5*), as mutations in both genes have been associated to rare diseases accompanied with compromised renal function [26]. Indeed, progressive muscle degeneration in Duchene's muscular dystrophy, caused by mutations in *DMD*, seems to be associated to subclinical kidney injury and the subsequent development of chronic kidney disease in paediatric patients [27]. Furthermore, and although insufficiently explored, neuromuscular impairment has been suggested to be part of the clinical spectrum associated with *CLDN19* mutations as intolerance to muscular exercise that persist after kidney transplant and normalization of magnesium levels has been described in a few patients [28]. On the other hand, the HPS5 protein is highly expressed in human renal tubules according to the Human Protein Atlas and *HPS5* knock-down resulted in a severe renal phenotype in a zebrafish model through a mechanism that involves intracellular accumulation of cell debris due to lysosomal impairment [26].

We also identified variants in genes not typically associated with the thick ascending limb of Henle where *CLDN16* and *CLDN19* function. Of particular interest are variants in *SLC17A1* and *SLC9A3*, which play important roles in other segments of the nephron. *SLC17A1* is a sodium-dependent phosphate transporter primarily expressed in the proximal tubule and involved in urate excretion [29]. *SLC9A3* is a sodium/hydrogen exchanger found in the proximal tubule and thick ascending limb of Henle, crucial for sodium reabsorption and pH regulation [30]. Although not directly linked to magnesium and calcium handling in FHHNC, variants in these transporters could potentially influence overall renal function and electrolyte balance. Their association with enhanced progression to renal failure in FHHNC patients suggests possible indirect effects on disease progression, perhaps through alterations in tubular function or acid-base homeostasis. In addition, variants in other members of the claudin family of proteins, *CLDN8* and *CLDN17*, are also associated to a more aggressive phenotype in our FHHNC cohort. Although information on the role of *CLDN17* in the human kidney is limited, it has been identified in the proximal tubules of both mice and humans [6]. Recently, it was reported to play a role in renal function by regulating serum electrolyte levels and tissue reactive oxygen species [31]. On the other hand, *CLDN8* has been implicated in the regulation of paracellular permeability to sodium and water in the distal nephron [32]. The identification of these variants highlights the complex interplay of various renal transporters and channels in maintaining kidney function. While the primary defect in FHHNC lies in the thick ascending limb, our findings suggest that genetic variations in other nephron segments may contribute to the variability in disease progression. This underscores the importance of considering the kidney as an integrated system, where dysfunction in one area can have far-reaching effects on overall renal health.

We acknowledge several important limitations in our study. First, the ultra-rare nature of FHHNC presents significant challenges in achieving robust statistical power. Our cohort, while comprehensive for Spain, remains small, which limits the generalizability of our findings and increases the risk of false positives. Replication studies in independent cohorts are mandatory. Our replication cohort, though valuable, has limitations. While we were able to recruit 15 additional patients, a significant achievement for an ultra-rare disease, and all prioritized candidate variants showed concordant results by a different technique in a larger patient cohort, we acknowledge this cannot be considered as a full replication study as this cohort lacked fast progressors, limiting our ability to fully validate the association of identified

variants with rapid disease progression. We recognize that a more comprehensive replication study including patients with the most severe renal phenotype, would strengthen our findings. In addition, we are aware that the multi-step methodology taken may appear as somehow circular. Our approach involved an initial identification of variants associated with renal progression through WES analysis and statistical criteria, followed by prioritization based on biological relevance to FHHNC traits. The subsequent overrepresentation analysis in the FAST group served as an independent validation step and, indeed, not all initially prioritized variants were overrepresented in this analysis, which further supports the robustness of our approach. This discrepancy allowed us to focus on variants consistently associated with faster renal failure progression across different analytical steps. Despite all these measures, the potential for unidentified confounding factors and the challenge of distinguishing true modifier effects from statistical noise in a small cohort remain significant concerns. While we have identified promising potential modifier variants in *NFU1*, *DMD*, *HPS5, CLDN8* and *CLDN17*, their functional impact on FHHNC progression towards kidney failure remains to be characterized. However, we would like to remark that even gathering information regarding the impact of the genetic variants on their respective protein function in cell and/or animal models, there would be still a huge gap to understand their contribution to such a complex phenotype as in FHHNC. Furthermore, since these variants have not been described to be pathogenic *per se*, in light of our results we can speculate that they interact together as part of the genetic context of patients with a more severe renal phenotype. How these variants are combined to interact with the FHHNC causative *CLDN19* mutation resulting in a particular renal phenotype is far more difficult to approach. Appropriate models for this disease, which arises from the thick ascending limb of Henle's loop of the nephron, simply do not exist and the knock-in mouse model carrying the *CLDN19* p.G20D causative mutation has not been generated either. Moreover, ethical constraints prevent renal biopsies in FHHNC patients, limiting our access to patient-specific tissue for direct molecular studies. Our reliance on public databases like GTEx and Human Kidney eQTL Atlas, while informative, may not fully capture the unique genetic context of FHHNC patients. In summary, while our findings are promising, they should be interpreted with caution. Further validation in additional cohorts or through functional studies will indeed be crucial to confirm the utility of these results before they can be considered for clinical decision-making. Despite these limitations, we believe our study provides valuable insights into potential genetic modifiers in FHHNC and lays the groundwork for future, larger-scale investigations in this challenging field.

In summary, we have applied a pipeline specifically tailored for an ultra-rare renal disease leading to the identification of several novel gene variants potentially associated with a more aggressive renal phenotype. Notably, the most promising candidates include variants that may affect the functional stability of critical cell organelles such as mitochondria and lysosomes. The orchestrated action of these organelles is key for cellular homeostasis and survival under pathological stress conditions [33] and may explain the variability of the clinical phenotype in FHHNC in which the endoplasmic reticulum (ER) is likely to be under high stress due to the reported retention of *CLDN19* p.G20D mutant protein [5]. Furthermore, the association of our candidate modifier genes with other diseases involving signs of kidney disease raises the interesting possibility that their involvement in renal failure progression may not be FHHNC-specific. Consequently, potential prognostic and therapeutic tools developed for these targets could offer broader applicability across various renal and extra-renal diseases. Finally, our approach can easily be customized to identify and study modifier genes for other renal rare diseases and can contribute to strengthen the interpretation of personalized exome sequencing approaches, providing, therefore, a valuable framework for the identification and understanding of phenotypic variability and disease risk.

## Methods

### Ethics statement

All patients, or their caregivers, provided written informed consent before participation in the study. This study was approved by the Ethics Committee of our institution, the Vall d'Hebron Hospital (PR(AMI)280/2015). Patient records were consulted only to obtain relevant patient data. The described research adhered to the Declaration of Helsinki.

### Cohort description

This study included a total cohort of 24 healthy controls and 45 patients with clinical and genetic confirmation of FHHNC and stratified according to our previously reported cutoff for the annual glomerular filtration rate (eGFR) decline estimation [12], as follows: $\geq 10$ mL/min/1.73m²/year for the fast renal progression group; $< 10$ mL/min/1.73m²/year for the moderate-renal progression group; no eGFR decline along time (stable renal function) for the slow-renal progression group. Genetic diagnosis and clinical phenotype is thoroughly described in our previous study in a subcohort of 30 patients [12].

### Study design and rationale

We sought to identify non-pathogenic gene variants influencing the phenotypic heterogeneity of FHHNC. Patients were stratified based on their annual eGFR decline, with the FAST group comprising those exhibiting the fastest renal function decline ($\geq 10$ mL/min/1.73m²/year) and the noFAST group encompassing all the other patients. The discovery cohort (n=30, recruited by January 2021) underwent whole exome sequencing (WES) and candidate prioritization based on an exhaustive disease knowledge-driven exploitation of data from public databases (Human Protein Atlas, GWAS catalog, GTEx, Human Kidney eQTL Atlas) and identifying risk variants by calculating odds ratios. Subsequently, during analysis of the discovery cohort data, an additional 15 patients were recruited for validation. Due to limited sample size, the validation analysis intentionally used the entire cohort (n=45) to maximize statistical power. This study describes the WES analysis pipeline and variant prioritization strategy employed to identify candidate genes potentially modifying the phenotype towards a more aggressive course of renal disease in FHHNC patients (see graphical abstract in Fig 1).

### Whole exome sequencing

**Library preparation and sequencing.** DNA from patients was obtained using the *Gentra Puregene Blood kit* (#158467, Qiagen), following manufacturers' instructions. Exome sequencing was performed at the Centro Nacional de Análisis Genómico (CNAG, Barcelona, Spain). Paired-end multiplex libraries were prepared according to manufacturer's instructions and enriched with the SureSelect Human All Exon v5 genome design exome kit (Agilent Technologies). Libraries were loaded to Illumina flowcells for cluster generation prior to producing 100 base read pairs on a NovaSeq6000 instrument following the Illumina protocol. Image analysis, base calling and quality scoring of the run were processed using the manufacturer's software Real Time Analysis (RTA 1.18.66.3) and followed by generation of FASTQ sequence files with the CASAVA software.

**Data analysis and variant calling.** Reads were mapped to Human reference genome v37 with decoy sequences (ftp://ftp.1000genomes.ebi.ac.uk/vol1/ftp/technical/reference/phase2_reference_assembly_sequence/hs37d5.fa.gz) with Burrows-Wheeler Alignment tool 0.7.17 (BWA-MEM) [34]. Alignment files containing only properly paired, uniquely mapping reads without duplicates were processed using Picard 2.20 [35] to add read groups and to remove

duplicates. The Genome Analysis Tool Kit (GATK 4.1.9.0) [36] was used for local realignment and base quality score recalibration. Variant calling was done using HaplotypeCaller from GATK. Functional annotations were added using SnpEff v5 with the GRCh37.75 database [37]. Variants were annotated with SnpSift v5 [38] using population frequencies, conservation scores and deleteriousness predictions from dbNSFPv4.1a [39]. Other sources of annotations, such as gnomAD, CADD and Clinvar were also used [40,41]. Statistical analysis was performed for variants of snpEff predicted impact moderate or high and with a coverage of at least 10 reads in one sample. Association tests were performed with Rvtests' SKAT-O [42]. Fisher Tests were additionally performed with R fisher.test package (Fisher's Exact Test for Count Data).

## Strategy for prioritization of candidate genes

The strategy we followed to select the most interesting candidate genes for further analysis was as follows:

1. A more stringent p-value threshold of <0.01 to minimize false positives

2. Genes expressed in the kidney (at gene and protein evidence) using the information obtained from the Human Protein Atlas.

Then we investigated whether significant genes obtained by SKAT-O had been previously associated with particular phenotype traits related to FHHNC. To do this, variant-trait associations from the GWAS catalogue [43] were used as a reference to assess the potential relevance of the candidate modifier genes identified by SKAT-O in our FHHNC cohort. Genes associated to FHHNC-related traits showing a P-value $< 1 \times 10^{-6}$ were selected and matched to our list of SKAT-O genes.

Finally, a second round of GWAS catalog – SKAT-O matching genes was performed including also genes not expressed in the kidney to explore potential influences from other tissues on FHHNC progression. Members of the claudin family of proteins were also included based on their well-established role in tight junction biology and potential involvement in kidney pathophysiology to explore broader mechanisms of tight junction dysfunction that may influence FHHNC outcomes.

## SNP validation by open arrays

A customized TaqMan OpenArray Genotyping Plate was designed including the final set of prioritized variants using predesigned TaqMan SNP Genotyping Assays (S7 Table). Each DNA sample was normalized to a concentration of 50 ng/μL, and 3 μL of DNA was mixed with the same volume of 2x TaqMan OpenArray Genotyping Master Mix and manually loaded into 384 well-plates. The QuantStudio 12K Flex OpenArray AccuFill System (Thermo Fisher Scientific) transferred the previously generated mix to the TaqMan OpenArray plate. The amplification was performed using the QuantStudio 12K Flex Real Time PCR System (Thermo Fisher Scientific) instrument, and the results were analysed using TaqMan Genotyper Software (Thermo Fisher Scientific).

## Functional profiling analysis

To gain further insights into the functional relevance of the genes identified by SKAT-O, several approaches and tools were used, as described below. The main goal was to group the prioritized candidate genes into pathways and biological processes to further explore their potential involvement in FHHNC phenotype modulation. The statistical analysis was

performed using the statistical language "R" (R version 4.2.0) [44] and libraries from the Bioconductor Project (www.bioconductor.org).

**g:Profiler analysis.** The version of g:Profiler used for this study was e107_54_p17_bf42210. The parameters for the enrichment analysis were as follows. *Homo sapiens* (human) was chosen as the organism for the analysis. GO analyses (GO molecular function (GO-MF), GO cellular component (GO-CC), and GO biological process (GO-BP)) were carried out sequentially. The statistical domain scope was used only for annotated genes and the significance threshold was the Benjamini and Hochberg FDR threshold at < 0.05.

**Overrepresentation analysis (ORA).** ORA of the selected genes was performed over the GO-BP database using clusterProfiler R package v4.4.1 [45]. The background distribution for the hypergeometric test was all the human genes that have annotation. Gene sets were obtained from the annotation package org.Hs.e.g.,db v.3.15.0. A list of the enriched terms sorted by P-value and filtered by an adjusted P-value <0.25 was obtained. Results were graphically represented as a dot plot of the top 15 terms enriched with an adjusted P-value <0.25. This less stringent cutoff was chosen to capture a broader range of potentially relevant biological processes. In this plot, the size of the dots relates to the number of genes/proteins in the data that belong to that pathway, the colour of the plot refers to significance level (P-value). The terms are ordered by P-value and Gene ratio, which is the ratio between the genes/proteins in the data that belong to that term and the total number of genes/proteins in the term.

**Summarization of ORA results.** To summarize the results obtained in the enrichment analysis over GO-BP, a clustering analysis was performed using the R/Bioconductor package simplifyEnrichment v1.6.1 [46]. This method allows to cluster terms using different similarity measures and clustering algorithms and visualize the results as Heatmaps. Here, terms with an adjusted P-value <0.25 were grouped based on Rel similarity measure and binary cut clustering algorithm [46]. This less stringent cutoff was chosen to capture a broader range of potentially relevant biological processes. Clusters were annotated with a word cloud representing the top 10 most frequent words found within term names in each cluster, and with the most representative term within the cluster, selected as the term with maximal number of SKAT-O genes and with the minimal P-value).

## Statistical analysis

The biostatistics analysis was focused on the identification of gene variants associated to the FAST renal progression phenotype. Prioritized modifier candidate genes were analysed by the exact Fisher test considering patients homozygous for the reference allele (R/R), heterozygous (R/A) and homozygous for the alternative allele (A/A) among the two phenotypes (FAST *vs.* noFAST) to identify those variants that distribute differently between both phenotypes. For significant variants, the odds ratios (OR) and 95% confidence intervals were calculated applying the Haldane and Anscombe correction [47], for the homozygous A/A or the heterozygous A/R genotypes compared to the homozygous R/R genotype.

To gain further insights on the association of prioritized variants with fast renal progression, we used eQTL data to check variant expression by genotype in 659 microdissected human kidney samples from the Human Kidney eQTL Atlas [48] and in 73 human kidney cortex tissues from the GTEx eQTL Dashboard. The data used for the analyses described in this manuscript were obtained from the GTEx Portal on 01/23. In addition, to find whether the distribution of all genotypes (A/A, A/R, or R/R) was the same for each group of subjects analysed (FAST, noFAST, GTEx) data was analysed by the exact Fisher's test.

The statistical analysis was performed using the statistical language "R" (R version 4.2.0 (2022-04-22), Copyright (C) 2022 The R Foundation for Statistical Computing). All p-values

are reported as calculated, allowing to interpret the strength of evidence for each finding. P-values were considered significant when p<0.05, except in the prioritization pipeline and enrichment analysis as explained above.

## Supporting information

**S1 Fig. Graphic representation of g:Profiler functional analysis of the candidate modifier genes obtained by SKAT-O.** The most significant results for Gene Ontology (GO) are shown. GO molecular function (GO:MF); GO biological process (GO:BP).
(TIF)

**S1 Table. Number of patients distributed in each group.**
(XLSX)

**S2 Table. Phenotypic traits related to FHHNC and matched SKAT-O genes to GWAS catalog associated variants.**
(XLSX)

**S3 Table. Significant single nucleotide variants associated to the fast renal progression phenotype.**
(XLSX)

**S4 Table. Prioritized phenotype modifier candidates and associated cluster and gene ontology biological process (GO-BP) from summORA analysis.**
(XLSX)

**S5 Table. Distribution of genotypes for each variant across the FHHNC phenotypes and the GTEx control group.**
(XLSX)

**S6 Table. Presence of risk alleles for the prioritized candidate SNVs showing aggregation in the FHHNC FAST renal progression phenotype.**
(XLSX)

**S7 Table. Selected variants and assays for validation by TaqMan® OpenArray® Genotyping platform.**
(XLSX)

## Acknowledgements

We would like thank the FHHNC patient advocacy group HIPOFAM (http://hipofam.org), patients and families for their valuable support and contribution to our research activity. We gratefully acknowledge the contributions of the medical doctors responsible for the recruitment and clinical management of the FHHNC national patient cohort, whose collaboration was essential to this study. We also thank Dr. Xavier de la Cruz for helpful discussions and advise. The Genotype-Tissue Expression (GTEx) Project was supported by the Common Fund of the Office of the Director of the National Institutes of Health, and by NCI, NHGRI, NHLBI, NIDA, NIMH, and NINDS.

## Author contributions

**Conceptualization:** Gema Ariceta, Anna Meseguer, Cristina Martinez.

**Data curation:** Monica Vall-Palomar, Alex Sánchez, Cristina Martinez.

**Formal analysis:** Monica Vall-Palomar, Julieta Torchia, Jordi Morata, Raul Tonda, Mireia Ferrer, Alex Sánchez, Cristina Martinez.

**Funding acquisition:** Gema Ariceta, Anna Meseguer.

**Investigation:** Monica Vall-Palomar, Julieta Torchia, Monica Durán, Gerard Cantero-Recasens, Gema Ariceta, Anna Meseguer, Cristina Martinez.

**Methodology:** Monica Vall-Palomar, Julieta Torchia, Monica Durán, Alex Sánchez, Anna Meseguer, Cristina Martinez.

**Project administration:** Anna Meseguer.

**Resources:** Gema Ariceta, Anna Meseguer.

**Supervision:** Anna Meseguer, Cristina Martinez.

**Validation:** Julieta Torchia, Cristina Martinez.

**Writing – original draft:** Cristina Martinez.

**Writing – review & editing:** Jordi Morata, Mireia Ferrer, Gerard Cantero-Recasens, Gema Ariceta, Anna Meseguer.

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
