## [Decision Letter · Decision Letter 0]

20 Sep 2024

Dear Dr Meseguer,

Thank you very much for submitting your Research Article entitled 'Identification of modifier gene variants overrepresented in familial hypomagnesemia with hypercalciuria and nephrocalcinosis patients with a more aggressive renal phenotype' to PLOS Genetics.

The manuscript was fully evaluated at the editorial level and by independent peer reviewers. The reviewers appreciated the attention to an important problem, but raised some substantial concerns about the current manuscript. Based on the reviews, we will not be able to accept this version of the manuscript, but we would be willing to review a much-revised version. We cannot, of course, promise publication at that time.

Specifically, reviewer 1 was concerned about the statistical analyses and the failure to correct for multiple testing. In addition, we were concerned about the lack of a validation cohort. Even though you did add new samples they could not actually validate as only one of the two phenotypes were added. We recognize that given the rarity of the disease this is difficult, but you will need to address both of these issues in any revised manuscript.

Finally, we would like to apologize for the delayed response, but it was unusually difficult to be able to have at least 3 expert reviewers, maybe because of the summer.

If you decide to revise the manuscript for further consideration at PLOS Genetics, please aim to resubmit within the next 60 days, unless it will take extra time to address the concerns of the reviewers, in which case we would appreciate an expected resubmission date by email to plosgenetics@plos.org.

If present, accompanying reviewer attachments are included with this email; please notify the journal office if any appear to be missing. They will also be available for download from the link below. You can use this link to log into the system when you are ready to submit a revised version, having first consulted our Submission Checklist .

PLOS has incorporated Similarity Check , powered by iThenticate, into its journal-wide submission system in order to screen submitted content for originality before publication. Each PLOS journal undertakes screening on a proportion of submitted articles. You will be contacted if needed following the screening process.

To resubmit, log into your Editorial Manager account and select the option 'Revise Submission' in the 'Submissions Needing Revision' folder.

We are sorry that we cannot be more positive about your manuscript at this stage. Please do not hesitate to contact us if you have any concerns or questions.

Yours sincerely,

Daniel Battle

Guest Editor

Scott Williams

Section Editor

PLOS Genetics

The paper has been revised by three experts and I wanted to apologize for the delayed decision, but finally I had enough feedback to make a decision. The data are unique and important and deserves publication after careful attention to the comments of reviewer 1 regarding statistics. Also please address the comments of the other reviewers.

Reviewer's Responses to Questions

**Comments to the Authors:**

Reviewer #1: The authors present a study on potential modifier variants for FHHNC. They perform WES in a cohort of 30 patients with FHHNC which they stratify by extremes with regards to kidney disease progression and identify 29870 variants in 11240 genes. Of these, 1068 genes were significantly associated with progression. They select from these 165 genes based on expression in the kidney and select further from these based on their representation in the GWAS catalogue as associated with phenotypic traits they consider relevant. They then add in genes from the GWAS catalogue associated with ocular impairment resulting in 44 candidate genes. They then add back in a few genes even though they are not expressed in the kidney (but with relevant GWAS catalogue association) and one because they like the name (it is a Claudin), ending up with 50 candidate gens containing 175 variants. These 175 variants were then individually tested for association with renal progression, of which 19 made the cut.

Major

1. Overall, this is a somewhat confusing paper. The selection of candidate genes and variants appears quite haphazard. A flowdiagramn would help to clarify the individual steps. Were the added in genes associated with ocular phenotype used for renal progression analysis? Analysed separately?

2. The statistics are quite confusing. P-values are considered significant sometimes at <0.05, sometimes <0.01, sometimes <0.25 and it is not clear how these levels were derived.

3. Most importantly: the fact that p-values were not corrected for multiple comparisons makes the whole statistics essentially useless. They state that they mitigate the risk of false-positives by using “state-of-the-art knowledge of the disease”, but why do statistics then in the first place? Who needs statistics if you have “state-of-the-art knowledge of the disease”?

4. Presumably, the p-value in the Fisher’s exact test for the individual variant association was also not corrected for multiple comparisons?

5. Line 203: I am very confused by this statement. These variants were selected because of their association with renal progression and then it is presented as a result that these variants are overrepresented in “fast progressers”? Isn’t that circular logic? Same for line 228.

6. Given the potential of the “winner’s curse”, confirmation of the results from the discovery cohort in a validation cohort is essential. The fact that the validation cohort contains no “fast progressers” makes it unsuitable: How can you validate risk factors for fast progression in a cohort without fast progression? The fact that the risk factors are absent does not exclude the winner’s curse.

minor

1. Abstract, line 38: not clear which gene the common variant is located in: CLDN16? Or 19?

2. Line 125: explain GATK

3. Line 127: explain SKAT-O

4. Line 131: explain GO-BP and GO-CC

Reviewer #2: The authors present a study on potential modifier variants for FHHNC. They perform WES in a cohort of 30 patients with FHHNC which they stratify be extremes with regards to kidney disease progression and identify 29870 variants in 11240 genes. Of these, 1068 genes were significantly associated with progression. They select from these 165 genes based on expression in the kidney and select further from these based on their representation in the GWAS catalogue as associated with phenotypic traits they consider relevant. They then add in genes from the GWAS catalogue associated with ocular impairment resulting in 44 candidate genes. They then add back in a few genes even though they are not expressed in the kidney (but with relevant GWAS catalogue association) and one because they like the name (it is a Claudin), ending up with 50 candidate gens containing 175 variants. These 175 variants were then individually tested for association with renal progression, of which 19 made the cut.

Major

1. Overall, this is a somewhat confusing paper. The selection of candidate genes and variants appears quite haphazard. A flowdiagramn would help to clarify the individual steps. Were the added in genes associated with ocular phenotype used for renal progression analysis? Analysed separately?

2. The statistics are quite confusing. P-values are considered significant sometimes at <0.05, sometimes <0.01, sometimes <0.25 and it is not clear how these levels were derived.

3. Most importantly: the fact that p-values were not corrected for multiple comparisons makes the whole statistics essentially useless. They state that they mitigate the risk of false-positives by using “state-of-the-art knowledge of the disease”, but why do statistics then in the first place? Who needs statistics if you have “state-of-the-art knowledge of the disease”?

4. Presumably, the p-value in the Fisher’s exact test for the individual variant association was also not corrected for multiple comparisons?

5. Line 203: I am very confused by this statement. You selected these variants because of their association with renal progression and then you present as a result that these variants are overrepresented in “fast progressers”? Isn’t that circular logic? Same for line 228.

6. Given the potential of the “winner’s curse”, confirmation of the results from the discovery cohort in a validation cohort is essential. The fact that the validation cohort contains no “fast progressers” makes it unsuitable: How can you validate risk factors for fast progression in a cohort without fast progression? The fact that the risk factors are absent does not exclude the winner’s curse.

minor

1. Abstract, line 38: not clear which gene the common variant is located in: 16? Or 19?

2. Line 125: explain GATK

3. Line 127: explain SKAT-O

4. Line 131: explain GO-BP and GO-CC

Reviewer #3: This is a very interesting and novel approach to the genetic risk factors of having a severe phenotype in patients with FHHNC. It is especially interesting taking into account that a high number of the patients of the cohort exhibit the same p.G20D variant, rendering a high genetic similarity that allows comparison between patients.

The main limitations of the study are, in my opinion, the low number of patients (specifically the low number of FAST patients), but this is understandable taking into account it is an ultra-rare disease. I find the authors discuss this point appropriately in their manuscript.

In addition, the genetic risk factors found in the FAST patients would need to be validated in other larger cohorts, as the validation cohort of 15 patients is probably not enough (again, especially the FAST patients). I find the authors also manifest in their manuscript this limitation point.

I have no other comments to the manuscript.

Reviewer #4: The manuscript by Vall-Palomar et al is titled “ Identification of modifier gene variants overrepresented in familial hypomagnesemia with hypercalciuria and nephrocalcinosis patients with a more aggressive renal phenotype” and studies individuals with familial hypomagnesemia with hypercalciuria and nephrocalcinosis (FHHNC) which is caused by loss-of-function mutations in CLDN16 and CLDN19. The study examines the phenotypic variability in a mutation which is the most prevalent mutation in Spain. Patients were stratified according to their estimated glomerular filtration rate annual decline. Whole exome sequencing was obtained to find candidate phenotype-modifier genes.

The authors report the identification of 19 putative modifier gene variants associated with a higher risk of developing a more aggressive renal phenotype.

The authors specifically report the identification of a panel of genetic variants within novel candidate modifier genes which could be utilized for estimating the risk of developing early kidney failure in this disease.

The studies are well designed and well executed. I have few minor concerns.

1. There are several major limitations with these studies, but the authors are aware of them and have discussed these limitations as they pertain to the impact of specific variants in NFU1, DMD, HPS5, CLDN8 and CLDN17 on faster decline in kidney function. They need to expand on these limitations.

2. The authors would like to discuss the possible role of variants in SLC17A1 and SLC9A3 and their association with enhanced progression to renal failure in FHHNC. This referee was not aware of any studies pointing to the localization of the urate transporter SLC17A1 in the thick ascending limb.

**Have all data underlying the figures and results presented in the manuscript been provided?**

Reviewer #1: **No: **

Reviewer #2: Yes

Reviewer #3: Yes

Reviewer #4: Yes

PLOS authors have the option to publish the peer review history of their article (what does this mean? ). If published, this will include your full peer review and any attached files.

**Do you want your identity to be public for this peer review?** For information about this choice, including consent withdrawal, please see our Privacy Policy .

Reviewer #1: No

Reviewer #2: No

Reviewer #3: No

Reviewer #4: No

---

## [Decision Letter · Decision Letter 1]

21 Nov 2024

PGENETICS-D-24-00610R1Identification of modifier gene variants overrepresented in familial hypomagnesemia with hypercalciuria and nephrocalcinosis patients with a more aggressive renal phenotypePLOS GeneticsDear Dr. Meseguer, Thank you for submitting your manuscript to PLOS Genetics. After careful consideration, we feel that it has merit but does not fully meet PLOS Genetics's publication criteria as it currently stands. Therefore, we invite you to submit a revised version of the manuscript that addresses the points raised during the review process. Please submit your revised manuscript within 30 days Dec 21 2024 11:59PM. If you will need more time than this to complete your revisions, please reply to this message or contact the journal office at plosgenetics@plos.org. Please include the following items when submitting your revised manuscript: * A rebuttal letter that responds to each point raised by the editor and reviewer(s). You should upload this letter as a separate file labeled 'Response to Reviewers '. This file does not need to include responses to formatting updates and technical items listed in the 'Journal Requirements' section below. * A marked-up copy of your manuscript that highlights changes made to the original version. You should upload this as a separate file labeled 'Revised Manuscript with Track Changes '. * An unmarked version of your revised paper without tracked changes. You should upload this as a separate file labeled 'Manuscript '. If you would like to make changes to your financial disclosure, competing interests statement, or data availability statement, please make these updates within the submission form at the time of resubmission. Guidelines for resubmitting your figure files are available below the reviewer comments at the end of this letter. We look forward to receiving your revised manuscript.Kind regards,

Daniel Batlle

Guest Editor

PLOS Genetics

Gregory CooperSection EditorPLOS GeneticsAimée DudleyEditor-in-ChiefPLOS Genetics  Anne GorielyEditor-in-ChiefPLOS Genetics**Additional Editor Comments:** The paper is much improved. One reviewer is still not satisfied, although states that the paper has improved. The main concern of this reviewer is the statistics which he / she states should be more stringent for this type of work which I do not agree, otherwise it would be impossible for ultra rare diseases to publish anything. The number of patients reported is the largest ever for this ultra rare disease (over 40)

The authors nevertheless should indicate as a limitation that clinical decisions should not be made based on the data provided and that further cohorts should be studied. They also have to clarify when they use the value of p <0.05 and/or p <0.01**Journal Requirements:**

1) We have noticed that you have uploaded Supporting Information files, but you have not included a complete list of legends. Please add a full list of legends for your Supporting Information files after the references list.

2) Please amend your detailed Financial Disclosure statement. This is published with the article. It must therefore be completed in full sentences and contain the exact wording you wish to be published. Please ensure that the funders and grant numbers match between the Financial Disclosure field and the Funding Information tab in your submission form. Note that the funders must be provided in the same order in both places as well.

**Reviewers' comments:**

Reviewer's Responses to Questions

**Comments to the Authors:**

Reviewer #2: The manuscript is improved, but the same methodological concerns remain, see below. I fully appreciate the difficulties of performing robust statistics in an ultrarare disorder, but does that really mean that we should burden these patients with potentially misleading and erroneous results? These results, once published, will almost certainly be picked up by patients, who may request testing for these results to inform important life decisions (“my unborn child has FHHNC and if it has risk variants for fast progression, I want to terminate that pregnancy”; or “ I am thinking of going to university now, but if I am carrying variants for fast progression, I probably won’t be able to conclude the studies because I have to start dialysis”). Their clinicians are then faced with trying to explain to these patients that these results are highly speculative, as the analysis bypassed robust statistical analysis. Or, worse, their clinicians may similarly take these results at face value as they are published in a respected peer-reviewed journal. Is the appropriate response to the problem of rare disorders really that we allow a 4.000-fold higher likelihood of false positive results?

1. Regarding comment 1: I did not want to imply that the authors added genes not identified in their initial association study. I simply wanted to summarise the candidate gene approach they took in that step, which, if I understand correctly, is still an appropriate summary. The key message here is that it is not an unbiased assessment of genetic variants, but a highly selected one, which obviously affects interpretation of the results. Did the authors decide on their selection strategy (which appears to be a] any gene expressed in the kidney and in selected categories of GWAS-catalogue, b] anything associated with eye and in selected categories of GWAS-catalogue, c] anything carrying the name claudin) a priori, or did they apply their “state-of-the art-knowledge” once they had the initial association results?

2. Line 139: Please detail, what percentile of predicted damaging variants the categories “moderate and high” would correspond to.

3. Lines 142 and 485: I am still confused by the statistics. In the rebuttal, the authors state that they now use FDR-corrected p-values, but in the results (line 142), they state that they still use p<0.05. And according to Methods (line 485), they now use a p-value of 0.01, instead of the previous 0.05: what is it now? And is a p<0.01 supposed to be the appropriate FDR correction when dealing with variants in ~20k genes? If so, based on which assumptions? Or is that an arbitrary level chosen by the authors? They state that it is “more stringent”: More stringent than what? An FDR-corrected p-value? The accepted corrected p-value for exome-wide gene-based analysis is 2.5x10^-6: the “more stringent” value of 0.01 is 4000 times larger, meaning that the risk that their results are false-positive is 4000 times larger than what is generally accepted in the scientific community! Do the authors really want to take responsibility for such an increased risk that their results are misleading?

4. Line 194: Again, a p-value of 0.05 is used, even though dealing with 175 variants: is that FDR-corrected?

Reviewer #3: I think this is an interesting and well-performed study. The authors correctly address what I find are the main limitations of the study: a limited number of patients and the lack of a sound validation cohort.

I also find the authors correctly address other reviewer’s concerns and have adapted the text to them.

Reviewer #4: The authors have adequately addressed the concerns that were raised by referees.

**Have all data underlying the figures and results presented in the manuscript been provided?**

Reviewer #2: Yes

Reviewer #3: Yes

Reviewer #4: Yes

PLOS authors have the option to publish the peer review history of their article (what does this mean? ). If published, this will include your full peer review and any attached files.

**Do you want your identity to be public for this peer review?** For information about this choice, including consent withdrawal, please see our Privacy Policy .

Reviewer #2: No

Reviewer #3: No

Reviewer #4: No

**Figure resubmission:**
---

## [Editor Report · Decision Letter 2]

8 Jan 2025

Dear Dr Mesaguer

We are pleased to inform you that your manuscript entitled "Identification of modifier gene variants overrepresented in familial hypomagnesemia with hypercalciuria and nephrocalcinosis patients with a more aggressive renal phenotype" has been editorially accepted for publication in PLOS Genetics. Congratulations!

Yours sincerely,

Daniel Batlle

Guest Editor

PLOS Genetics

Gregory Cooper

Section Editor

PLOS Genetics

Aimée Dudley

Editor-in-Chief

PLOS Genetics

Anne Goriely

Editor-in-Chief

PLOS Genetics

Comments from the reviewers (if applicable):

**Data Deposition**

http://datadryad.org/submit?journalID=pgenetics&manu=PGENETICS-D-24-00610R2

**Press Queries**

---

## [Editor Report · Acceptance letter]

PGENETICS-D-24-00610R2

Identification of modifier gene variants overrepresented in familial hypomagnesemia with hypercalciuria and nephrocalcinosis patients with a more aggressive renal phenotype

Dear Dr Meseguer,

We are pleased to inform you that your manuscript entitled "Identification of modifier gene variants overrepresented in familial hypomagnesemia with hypercalciuria and nephrocalcinosis patients with a more aggressive renal phenotype" has been formally accepted for publication in PLOS Genetics! Your manuscript is now with our production department and you will be notified of the publication date in due course.

With kind regards,

Zsofia Freund

PLOS Genetics

On behalf of:
